# Evaluation of a Smart Knee Brace for Range of Motion and Velocity Monitoring during Rehabilitation Exercises and an Exergame

**DOI:** 10.3390/s22249965

**Published:** 2022-12-17

**Authors:** Michelle Riffitts, Harold Cook, Michael McClincy, Kevin Bell

**Affiliations:** 1Department of Bioengineering, University of Pittsburgh, Pittsburgh, PA 15206, USA; 2Department of Orthopaedic Surgery, University of Pittsburgh, Pittsburgh, PA 15206, USA

**Keywords:** wearable sensors technology, ACL injury, exergame, rehabilitation, range of motion monitoring

## Abstract

Anterior cruciate ligament (ACL) injuries often require a lengthy duration of rehabilitation for patients to return to their prior level of function. Adherence to rehabilitation during this prolonged period can be subpar due to the treatment duration and poor adherence to home exercises. This work evaluates whether a smart instrumented knee brace system is capable of monitoring knee range of motion and velocity during a series of common knee rehabilitation exercises and an exergame. A total of 15 healthy participants completed a series of common knee rehabilitation exercises and played an exergame while wearing a smart instrumented knee brace. The range of motion (ROM) and velocity of the knee recorded by the knee brace was compared to a reference optoelectronic system. The results show good agreement between the knee brace system and the reference system for all exercises performed. Participants were able to quickly learn how to play the exergame and scored well within the game. The system investigated in this study has the potential to allow rehabilitation to occur outside of the clinic with the use of remote monitoring, and improve adherence and outcomes through the use of an exergame.

## 1. Introduction

Anterior cruciate ligament (ACL) injuries are an increasingly common injury in the United States, with over 200,000 injuries occurring annually [1,2]. Younger patients in particular are at risk of ACL injury due to the non-contact mechanism of injury that is commonly found at high levels of competition in this age group [3]. ACL reconstruction (ACL-R) is usually the treatment of choice for these patients as the surgical procedure restores knee laxity and allows patients to return to pivoting sports [1]. Following surgery, patients undergo a lengthy postoperative rehabilitation period, with the goals of restoring full knee range of motion, preventing muscle hypotrophy, decreasing pain and swelling, and avoiding unnecessary stress to the repair site. Rehabilitation typically runs from the first week after surgery for 6–9 months postoperatively and typically involves 2–3 weekly visits to an outpatient physical therapy clinic [1,2].

The prolonged rehabilitation period allows for the monitoring of on-field exercises, along with the inclusion of agility and plyometric drills, which is important as lingering deficits can lead to re-injury, such as a graft rupture or contralateral ACL rupture [4]. This duration of supervised physical therapy following ACL-R has been associated with improved biomechanics and patient reported outcomes [5,6,7]. The rehabilitation period is often expensive and can result in patients exhausting their insurance allotted visits and cutting treatment short [8]. Treatment that allows for initial rehabilitation to be performed outside of the clinic and monitored remotely could allow for in-person visits to be saved for later in the course of care and for higher-level rehabilitation. Home exercise programs (HEPs) are a group of selected exercises that a physical therapist assigns a patient to complete on their own outside of the clinic [9]. HEP performance is not always formally monitored by the prescribing physical therapist, and shifting to a remotely monitored HEP system could allow for initial treatment outside of the clinic, as well as improve adherence [10]. Non-adherence to HEPs has been linked to stress, negative feelings, and a lack of positive feedback [6,11,12,13]. Adherence to rehabilitation treatment has been positively associated with improved patient outcomes [14,15].

Gamification has the potential to increase treatment adherence. Defined as the use of game design elements in non-game contexts [16], gamification has resulted in increased rehabilitation engagement [17], high compliance rates [18], and improved outcomes [19]. The goal of gamification is to influence behavior and motivation through the use of game mechanics (quests, points, leaderboards, badges) in an effort to improve engagement [20,21,22]. A recent randomized controlled trial used exergames (video games that require physical exertion or movements that include strength, balance, and flexibility activities) [23] and remote monitoring as a form of rehabilitation following arthroscopic shoulder surgery. Compared to the control conventional care rehabilitation group, the exergame and remote monitoring group demonstrated equal post-operative outcomes, indicating exergames and remote monitoring can be used effectively as a form of rehabilitation [24]. In a previous study, adolescent ACL-R patients expressed interest in the development of a gamified home exercise tracking system to use during their rehabilitation [10].

Gamification and the inclusion of exergames in post-operative knee surgery and ACL-R rehabilitation has been previously explored [25,26,27,28,29]. Fung et al. performed a preliminary randomized control trial using a Wii Fit in conjunction with traditional outpatient physical therapy following total knee replacement surgery. They found the exergame group to have equivalent outcomes to the group that received standard outpatient physical therapy [25]. Ficklscherer et al. conducted a randomized controlled study with post-operative knee surgery patients (TKR and ACL-R) to assess the feasibility and safety of using a Nintendo Wii in rehabilitation following knee surgery. They found no difference between the group that utilized the Wii in their treatment and those that received standard rehabilitation. Importantly, they reported that using the Wii in treatment was highly accepted by patients, concluding that utilizing an exergame-based rehabilitation could increase motivation and rehabilitation adherence, thereby having a positive sustainable effect on the healing process [26]. Baltaci et al. used a Nintendo Wii rehabilitation intervention in an experimental group of ACL-R patients to determine if using the Wii resulted in a difference in functional performance outcomes. They also found no difference between the exergame and control groups [27]. Ahmed et al. had an experimental group of ACL-R patients use a Wii Fit in their rehabilitation for 8 weeks. Following this 8 week treatment period, the experimental group demonstrated improved knee proprioception compared to the control group that received standard rehabilitation [28]. Clausen et al. developed a serious gaming intervention with a strength monitoring unit where patients played the game for 5 min at a time. Compared to a control standard rehabilitation group, the serious games group demonstrated improved maximum strength after a 6-week time period [29]. In these studies, both custom [26,29] and standard commercially available games [25,27,28] were used. Custom games offer the benefit of being personalized to the patient and their abilities, while standard games are readily available but may not have been medically developed and could have a potential risk of injury [26]. In these studies, good compliance was observed in the gaming group [29], and high acceptance rates of the gamified treatment were reported via satisfaction surveys [25,26].

These studies demonstrate that incorporating exergames into post-surgical knee rehabilitation is feasible and at least as effective as traditional rehabilitation methods. Additionally, the studies show that gamified treatment is accepted by patients and can result in high compliance rates. In this study, we introduce a novel instrumented smart knee brace that allows for knee range of motion (ROM) to be remotely monitored during common knee rehabilitation exercises and an exergame. Knee ROM is typically limited following ACL-R and restoring full ROM is important to fully regain gait and function [2,30]. The overall purpose of this work is to determine if an instrumented smart knee brace is capable of accurately tracking knee ROM and velocity during common knee rehabilitation exercises and an exergame.

## 2. Materials and Methods

### 2.1. Participants

Participants with no history of knee injury or disability were recruited as a convenience sample for this study. Inclusion criteria included being at least 18 years of age, having a healthy knee, and being able to perform functional movements of the knee. Exclusion criteria included: having a history of knee injury (defined as received medical care after an acute knee injury), having a history of knee disability (defined as having sought medical care to treat chronic knee pain or problems), a history of knee surgery, and having exercise restrictions. IRB approval was completed before the performance of any research activities. Participants were recruited through the University of Pittsburgh’s Human Movement and Balance Laboratory, and were compensated $25 for their time.

### 2.2. Device Description

A smart instrumented knee brace was constructed from a standard post-operative knee brace (X-ROM, Universal Size, DJO Global, Lewisville, TX, USA), a potentiometer (SV01A103AEA01R00, Noyito Technologies, Longhutang, Xinbei District Changzhou China) and a wireless Bluetooth transmitter (ESP-WROOM-32 ESP32 ESP-32S WiFi + Bluetooth Dev Board, HiLetgo, Shenzhen, China) (Figure 1A,B).

With this instrumentation, the knee brace is able to track knee joint range of motion (ROM) in real time and transmit this data wirelessly. To assess the accuracy of the knee brace motion tracking system, reflective markers were attached to the brace and a standard video-based motion capture system OptiTrack, (Optihub 2, NaturalPoint, Inc., Corvallis, OR, USA) was used as a gold standard comparison. Reflective markers were attached to a rigid body base (Rigid Body Marker Base, NaturalPoint, Inc., Corvallis, OR, USA) and secured to the outside of the knee brace at thigh, knee, and shank (Figure 2). One rigid body base with 4 reflective markers was placed on the thigh upright, one rigid body base with 4 reflective markers was placed at the joint of the brace (knee joint) and one rigid body base with 5 reflective markers was placed on the shank upright.

### 2.3. Experimental Procedures

#### 2.3.1. Calibration Procedure

Participants wore the instrumented knee brace on their right leg. To begin, the knee brace was locked at 90 degrees of flexion and participants were instructed to flex their knee until the brace provided a mechanical hard stop, this process was repeated 3 times. The brace was then unlocked to allow for full range of motion.

#### 2.3.2. Rehabilitation Exercises

Participants were instructed on how to complete several common knee rehabilitation exercises while wearing the knee brace. Standardized verbal instructions were provided on how to complete the exercises and participants were given the opportunity to ask questions or ask for clarification on the exercise technique. Participants completed 5 repetitions each of: maximum active knee flexion, maximum active knee extension, short arc quad (SAQ), straight leg raise (SLR), standing hamstring curl, and minisquat, ending with repeating the maximum active knee flexion and maximum active knee extension exercise [1,2,31]. During the exercises, knee joint ROM data was collected from both the knee brace and the OptiTrack systems.

#### 2.3.3. Exergame

After the completion of the knee rehabilitation exercises, participants played an exergame while wearing the knee brace. The exergame was custom coded in Python and allowed for continuous knee joint monitoring with the knee brace. The goal of the game is to use knee motion to control a small ball onscreen and navigate through varying gaps between vertical obstacles. The ball moves across the screen at a constant rate. Flexing the knee moves the ball up on the screen and extending the knee moves the ball down on the screen (Figure 3). For each obstacle that is successfully avoided players earn a point. Obstacles of different heights appear on screen at a predetermined standardized pace and the game automatically ends after the navigation of 20 obstacles. For the duration of the exergame, knee joint ROM data was collected from both the knee brace and the OptiTrack systems.

The game was developed with the intent of having participants move their knee through a controlled range of motion, reaching an angle of deep knee flexion in order to mimic performing a heel slide exercise. Participants controlled the ball on screen by moving their knee through a range of motion of 10–110 degrees of flexion. The game design was intended to have participants complete multiple repetitions of the heel slide exercise in a controlled way. The different height obstacles and the gaps between them were used to set the pace of the game and exercise. Game pace was the same for all participants and was set to allow for the completion of the heel slide exercise in a controlled manner. Points were awarded for the successful avoidance of an obstacle, encouraging participants to play the game well and therefore complete the exercise correctly.

### 2.4. Data Processing and Analysis

ROM data for the knee brace system was recorded directly from the potentiometer embedded in the knee brace for both the exercise and exergame portions of the study. ROM for the OptiTrack system was calculated with a custom MATLAB program (MATLAB, The MathWorks Inc., Natick, MA, USA) for both portions of the study. Rigid bodies for the thigh markers and shank markers were created and the position of the rigid body in space was extracted directly from the data recorded in the OptiTrack system. A zero- and ninety-degree calibration was performed prior to data collection for each participant. The knee brace was locked to 0 degrees of flexion and the position of the reflective markers in space was recorded with the OptiTrack system. The process was repeated with the brace locked at 90 degrees of flexion. Additional reflective markers were placed at the center of the medial knee joint, and on the anterior thigh and shank. A first axis (r1) was defined running from the medial knee joint to the lateral knee joint of the knee brace. A second axis (r2) was defined running from the anterior thigh to the anterior shank. A third axis, r3 was defined by taking the cross product of r1 and r2 (Equation (1)).
r3 = r1 × r2,(1)

This third axis was then crossed with one of the original axes to create three perpendicular axes that are all perpendicular to each other (Equation (2)).
r4 = r3 × r1,(2)

All axes were then normalized and combined into a 3 × 3 matrix. The matrix was converted to a quaternion representing the zero position of the brace (Q_0_). The quaternions representing the position of the thigh (Q_thigh_raw_) and shank (Q_shank_raw_) rigid bodies were extracted directly from the raw data recorded by the OpiTrack system and normalized. Quaternion multiplication was performed between the conjugate of the zero matrix and thigh rigid body (Equation (3)) and then the conjugate of the zero matrix and the shank rigid body (Equation (4)).
Q_thigh_ = conj(Q_0_) × Q_thigh_raw_,(3)
Q_shank_ = conj(Q_0_) × Q_shank_raw_,(4)

Equations (3) and (4) were multiplied to find the knee joint angle (Equation (5)). The resulting quaternion was decomposed into rotation angles with rotation order ‘XYZ’.
Q_knee_ = conj(Q_thigh_) × Q_shank_,(5)

The differences between the knee joint angle from the OptiTrack system at the zero- and ninety-degree calibration was averaged and added to the knee joint angle recorded by the OptiTrack system for the duration of the trial.

For the exercises, maximum and minimum ROM values (maximum flexion and extension values) associated with exercise performance were identified for each system. Corresponding values were compared with a root mean square error (RMSE) to quantify the error of the knee brace compared to the OptiTrack gold standard. For each exercise the inter-reliability between the two systems was estimated with an interclass correlation coefficient (ICC). The ICC was calculated by averaging the five different trials of the same exercise and then calculating the ICC based on these average values from linear mixed models with a fixed rater effect and random subject effect. ICC were rated as excellent (0.9–1), good (0.74–0.89), moderate (0.4–0.73), and poor (0–0.39) [32]. The differences between the OptiTrack and knee brace systems were displayed graphically with Bland–Altman plots. Velocity data for each exercise was calculated by taking the first derivative of the ROM data for each exercise. Local minima and maxima were identified for each system and corresponding data points were compared via RMSE. The statistical analysis for this paper was generated with SAS software, version 9.4 (SAS^®^ 9.4, SAS Institute Inc., 2013, Cary, NC, USA).

For the duration of the exergame trial, local minima and maxima ROM values were identified in both the knee brace and the OptiTrack data. Corresponding values were compared with a root mean square error (RMSE) to measure the error of the knee brace compared to the OptiTrack gold standard. Velocity data was calculated by taking the first derivative of the ROM data for each system. Local minima and maxima points were identified in the velocity data and compared via RMSE.

## 3. Results

### 3.1. Participants

Fifteen healthy participants with no history of knee injury were recruited for this study. Participants had an average age of 22.7 years (standard deviation = 1.9 years) and 11/15 participants were female. Participants had an average height of 165.4 cm (standard deviation = 13.0 cm) and an average weight of 66.2 kg (standard deviation = 10.5 kg). This study complies with the regulations of the Declaration of Helsinki and was approved by the University of Pittsburgh’s Institutional Review Board (STUDY20110132). All participants completed an informed consent process prior to participation.

### 3.2. Agreement between Knee Brace and OptiTrack System for Knee Exercises

#### 3.2.1. Range of Motion

The difference between the knee brace and the OptiTrack system for each exercise can be visualized in the Bland–Altman plots in Figure 4. The mean difference for each exercise can be seen in Table 1. From the Bland–Altman plots, the average differences between the knee brace and the OptiTrack systems were relatively low, with the largest difference being for the hamstring curl (1.09°) and minisquat (1.07°) exercises. The upper and lower limits of agreement were largest for the minisquat exercise (upper limit 7.18°, lower limit −5.04°) and SLR (upper limit 5.48°, lower limit −6.44°).

The ICC was highest for the minisquat exercise (0.98) and lowest for the SLR exercise (0.92). ICCs for all exercises can be seen in Table 2. RMSE values ranged from 1.62° to 3.07° with the SAQ exercise having the smallest value and the minisquat exercise having the largest value. RMSEs for all exercises can be seen in Table 2.

#### 3.2.2. Velocity

The RMSE values for the velocity ranged from 2.95°/s to 10.41°/s for the exercises. The SLR exercise had the smallest RMSE, and the hamstring curl exercise the largest. RMSE for all exercises can be seen in Table 3.

### 3.3. Exergame Results

RMSE of each participant’s local minima and maxima for ROM and velocity can be seen in Table 4, along with participant’s individual game scores. RMSEs between the knee brace and the OptiTrack systems for ROM ranged from 0.70° to 4.53° across participants. The total RMSE (RMSE of all 15 participants’ ROM local minima and maxima) was 2.30°. For velocity measurements, individual RMSEs ranged between 2.51°/s and 7.45°/s. The total RMSE for all velocity local minima and maxima was 5.06°/s. The average game score was 18.73 (out of a maximum 20).

## 4. Discussion

The goal of this study was to determine if a smart instrumented knee brace is capable of accurately tracking knee ROM and velocity during common knee rehabilitation exercises and an exergame through the comparison to a gold standard. The results show that the knee brace’s ROM and velocity measurement was in agreement with the optoelectronic motion tracking standard for a variety of exercises and for the duration of the exergame. Participants scored well in the exergame, showing success within a game controlled by knee ROM is possible.

For the knee rehabilitation exercises, the ICC comparing the knee brace’s recorded ROM to the corresponding OptiTrack values was excellent for all exercises, indicating a high inter-rater reliability between the two systems. The largest RMSE for an exercise was 3.07° for the minisquat exercise, which is still consistent with other similar knee brace motion tracking systems [33,34]. The RMSE values for the velocity data was higher compared to the RMSE values for that exercise’s ROM. The larger RMSE value for velocity could be related to the larger scale associated with velocity and the additional noise introduced via the derivation process. The hamstring curl exercise’s velocity RMSE was the highest, 10.41°/s, meaning this exercise had the most error between systems. The hamstring curl exercise was the only standing open-chain exercise performed, and additional error could have been introduced based on the movement of the optical markers when participants flexed their knee in this position.

The ICC value for ROM for both the first and second set of maximum flexion and extension exercises was excellent. The RMSEs for ROM for the first and second set of the maximum flexion exercise were within 0.5° of each other. The RMSEs for ROM for the first and second set of the maximum extension exercise were within 1°. The difference in the RMSEs for velocity of the first and second set of the maximum flexion exercise was within 2°/s. For the maximum extension exercise, the velocity RMSEs were within 1°/s of each other. The first and second set of these exercises were performed a few minutes apart, indicating that after several minutes the knee brace system is capable of recording accurate motion and velocity over a period of time.

For the exergame, individual participant RMSEs for ROM ranged from 0.73° to 4.53° and total RMSE from all participant data was 2.30°. For velocity, individual participant RMSEs ranged from 2.51°/s to 7.45°/s. The RMSE for all velocity local minima and maxima for all participants for the duration of the exergame was 5.06°/s. Once again, the RMSEs for ROM are consistent with similar systems [33,34] and the velocity RMSE showed more error. For the exergame, the average score was 18.73 out of 20 and several participants were able to score a perfect 20 by avoiding all 20 pipes. Participants were able to score well within the game and were able to quicky understand how to control the game with their knee motion.

There are several limitations to this study. RMSEs for both ROM and velocity, for both the standard rehabilitation exercises and the exergame, varied participant to participant. This result could indicate a difference in the knee brace’s ability to monitor and record knee motion and velocity, potentially related to the anthropometric properties of the participant’s leg or brace fit. The study was performed over a relatively short period of time, and the brace was calibrated prior to use by every participant. Even though the results from the first and second set of the maximum flexion and extension exercises were similar, the knee brace’s capability to accurately monitor motion over a longer period of time remains unknown. Additionally, participants were instructed on how to perform a certain set of exercises. The first several exercises (maximum flexion, maximum extension, SAQ, SLR) were performed in a supine position. The participant then stood up to complete the minisquat and hamstring curl exercises before returning to a supine position for the remaining exercises (maximum flexion and maximum extension) and the playing of the exergame. For this study, the participant only changed position twice while wearing the knee brace. Due to this limited number of transfers and minimal walking, it is not known how accurately the brace is capable of tracking knee motion and velocity during or after walking, transfers, or other activities of daily living, as errors could potentially result from movement or slippage of the brace.

Future work will explore the knee brace’s ability to monitor knee motion and velocity during activities of daily living and functional activities such as walking. Additionally, further effort will be put into the exploration and development of exergames, in order to make the game interesting and motivating enough to demand continued engagement throughout the rehabilitation duration. Future steps may include the addition of exergames for different rehabilitation exercises, along with the capability to compile scores from different games into a conglomerate score that can be included in a leaderboard. Future work should also include assessments on participants with altered knee mechanics, such as patients post-op ACL-R.

This study examined if a smart instrumented knee brace is capable of accurately measuring knee range of motion and movement velocity during a series of common knee rehabilitation exercises and a custom exergame. This study serves as the first step in developing a rehabilitation system that allows for exercise remote monitoring outside of the clinic. This system will allow for monitoring and rehabilitation to be performed remotely outside of the clinic, potentially reducing the financial burden following ACL-R and allowing for the completion of the duration of rehabilitation as the initial treatment could be performed remotely. The exergame system has the potential to improve treatment adherence through the use of an exergame.

## 5. Conclusions

This study found that an smart instrumented knee brace is capable of accurately monitoring knee range of motion and velocity during common knee rehabilitation exercises and a custom exergame. Participants were able to quickly learn how to play an exergame controlled by their knee motion. A system such as the one investigated could allow for remote rehabilitation to take place outside of the clinic and has the potential to improve adherence and outcomes through the incorporation of an exergame.

## Figures and Tables

**Figure 1 sensors-22-09965-f001:**
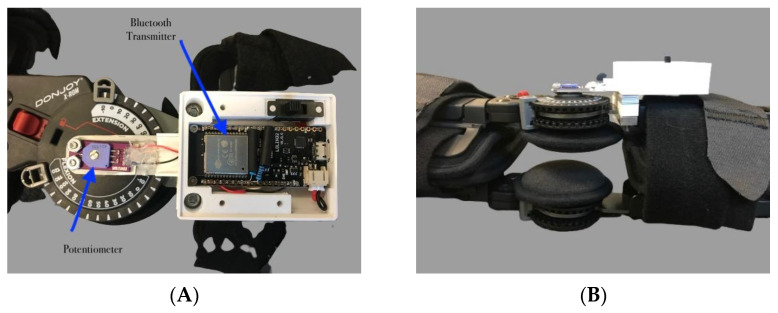
(**A**) Instrumented knee brace with potentiometer and Bluetooth transmitter. (**B**) Profile view of instrumentation in housing.

**Figure 2 sensors-22-09965-f002:**
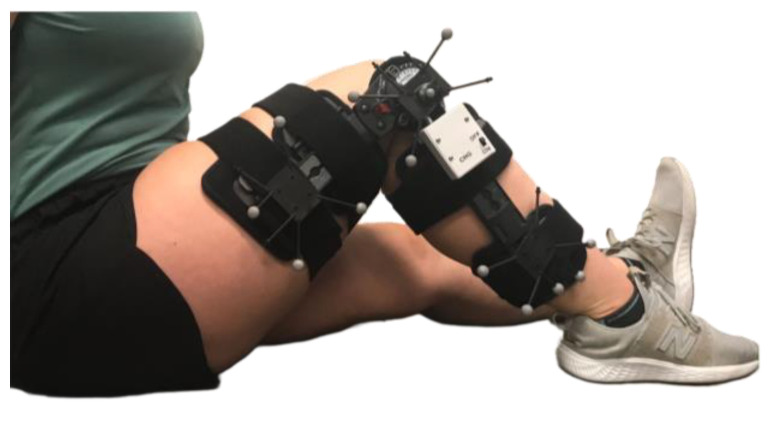
Instrumented knee brace with rigid body bases and reflective markers attached at the thigh, knee, and shank.

**Figure 3 sensors-22-09965-f003:**
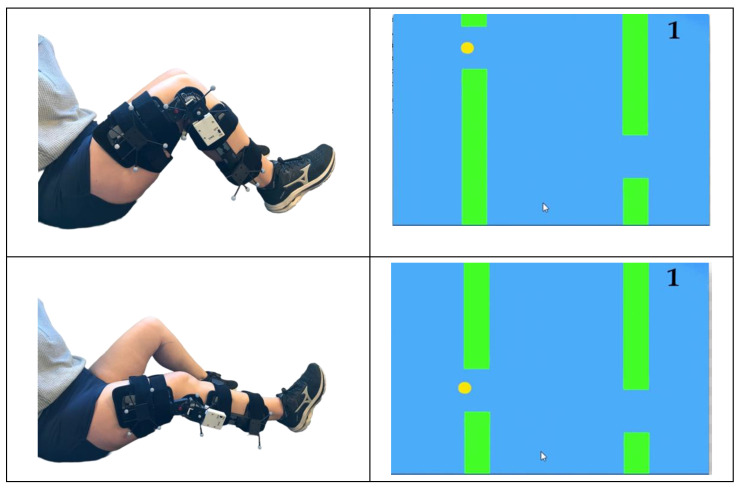
Exergame set up. Yellow ball onscreen is controlled by knee motion (flexion/extension) while wearing the instrumented knee brace. The ball moves across the screen at a constant pace and players flex/extend their knee to move the ball up and down to avoid the obstacles. For each obstacle that is avoided, players earn a point. The score is displayed in upper right-hand corner.

**Figure 4 sensors-22-09965-f004:**
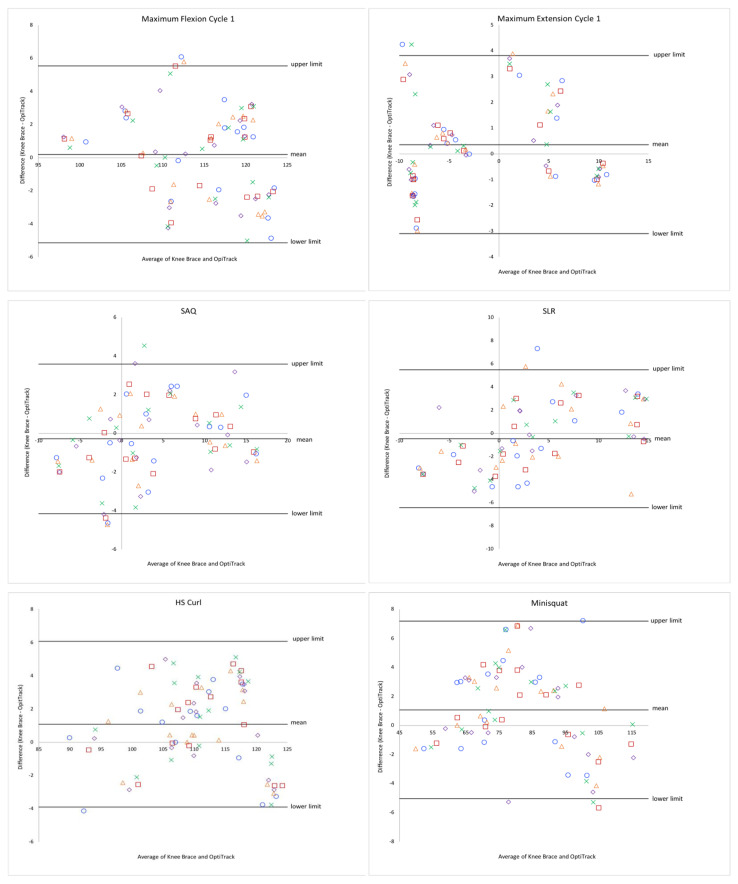
Bland–Altman plots for the range of motion of each exercise for 5 repetitions.

**Table 1 sensors-22-09965-t001:** Average difference between knee brace and OptiTrack systems for each exercise from Bland–Altman plot.

Exercise	Average Difference (°)	Lower Limit (°)	Upper Limit (°)
Maximum Flexion	0.19	−5.14	5.53
Maximum Extension	0.36	−3.10	3.82
SAQ	−0.28	−4.16	3.59
SLR	−0.48	−6.44	5.48
Hamstring Curl	1.09	−3.90	6.07
Minisquat	1.07	−5.04	7.18
Maximum Flexion 2	−0.13	−4.96	4.70
Maximum Extension 2	−0.74	−5.29	3.81

**Table 2 sensors-22-09965-t002:** Inter-rater reliability between the Knee Brace and OptiTrack Systems for each exercise for ROM measurements.

Exercise	ICC	RMSE (°)
Maximum Flexion	0.93	2.61
Maximum Extension	0.97	1.75
SAQ	0.97	1.62
SLR	0.94	2.62
Hamstring Curl	0.95	2.67
Minisquat	0.98	3.07
Maximum Flexion 2	0.95	2.35
Maximum Extension 2	0.95	2.35

**Table 3 sensors-22-09965-t003:** Inter-rater reliability between the Knee Brace and OptiTrack Systems for each exercise for velocity measurements.

Exercise	RMSE (°/s)
Maximum Flexion	7.42
Maximum Extension	3.86
SAQ	5.03
SLR	2.95
Hamstring Curl	10.41
Minisquat	3.29
Maximum Flexion 2	8.94
Maximum Extension 2	3.29

**Table 4 sensors-22-09965-t004:** Summary of exergame results, including ROM and velocity RMSE and game score broken down by participant.

Participant	ROM RMSE (°)	Velocity RMSE (°/s)	Score (Pipes Avoided)
1	4.53	3.52	18
2	0.70	4.79	19
3	3.07	5.79	20
4	1.44	6.94	18
5	1.65	5.43	17
6	1.37	5.49	18
7	2.36	4.56	19
8	1.92	7.45	19
9	2.37	6.00	20
10	2.87	2.66	17
11	1.95	3.52	19
12	0.73	4.56	20
13	3.12	5.75	18
14	2.13	3.47	20
15	1.83	2.51	19

## Data Availability

The data presented in this study is available on request from the corresponding author.

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
