# Peer review of "Evaluation of a Smart Knee Brace for Range of Motion and Velocity Monitoring during Rehabilitation Exercises and an Exergame"

_sensors, 2022, doi:10.3390/s22249965_

Round 1

Reviewer 1 Report

Thank you for the opportunity to review this work. It was a pleasure as the manuscript was well written, the methods were nicely outlined and the analysis was thorough. This work is important as few rehabilitation researchers have sought to find an association between rehabilitation exercises intended and a quantifiable point-system in a game.

Minor typos:

line 67: “randomizeD”

line 249: A rather than AN smart

line 262: additional noise - no OF required

Major comments:

Introduction: 

  1. I would like the authors to include one or two sentences to clarify a point of confusion in most fields about gamification. As the authors state lines 54-56, "gamification is the use of game design elements”. Most people believe that using a game-like interface (without intention and rationale) on a health-related activity means gamifying it. Gamification is the deliberate and purposeful use of game elements (e.g., competition, leader boards) to engage and motivate. The audience of this article should understand that gamification isn’t simply a nice visual that a developer created for a health-related activity; rather, it is the understanding of what motivates a patient group, an understanding of the activity requested of them, and the term of intended engagement with the health activity (e.g., workout term) then leveraging elements typically associated with games to promote adherence to these health activities.

  1. The overview of prior literature (like 65) could benefit from more critical synthesis (e.g., what were the Wii games used and do the present authors think it mattered? How did the cited authors determine that Wii treatment was “highly accepted” and do the authors of this manuscript have any comments on these findings? How is motivation increased or measured?) It would be nice for the present authors to offer their take on these findings as a whole.
  Materials/ Methods: 1) If in alignment with the Journal's specifications, I recommend moving the content of the "Participants" section to the "Results" whereby the authors let the audience know who participated in the study. In the methods, related to Participants, I would expect to see who the researchers intended to study and why. For example, why healthy participants, what were the inclusion/ exclusion criteria with justification, why 15 participants, was this the intended/ calculated sample size, did anyone decline participation, where was recruitment made and was there compensation for participation etc.
2) The procedure and instrumentation were well outlined and clearly described. Thank you. However, the authors should describe the decision making process behind the visual biofeedback provided by the "game". Why is this considered a game? Why a ball, and why gaps and obstacles? (These are important factors to outline when discussing gamification). How was the pace pre-determined? Is it the same for everyone, or calibrated based on the participant’s performance? For instance, the authors used a points-system - can the authors comment on the intention behind points?

Results and Discussion are clearly outlined.

Reviewer 2 Report

This work is well within the scope of Sensors, and it may be of interest to most of the readers of this journal. The proposed system could assist rehabilitation outside of the clinic with the use of remote monitoring and exergames. It shows an introductory background material sufficient for someone not an expert in this area to understand the context and significance of this work, with good references to follow. My only concern regarding this study, is that a small group of young and healthy volunteers were used to perform the tests.

Specific comments

Introduction 

P2, L79: ‘resulted in a difference in a difference’ Please correct. 

P9, L261-263: ‘The higher RMSE value for velocity could be related to the larger scale associated with velocity and the additional of noise introduced via the derivation process. The hamstring curl exercise’s velocity RMSE was the highest, 10.41°/second, meaning this exercise had the highest error between systems.’ Please consider the use of the words Higher, highest etc. since if the participants sample could be larger these values could have changed. 

P10, L286-289: ‘There are several limitations to this study. RMSEs for both ROM and velocity, for both the standard rehabilitation exercises and the exergame varied participant to participant. This result could indicate a difference in the knee brace’s ability to monitor andrecord knee motion and velocity, potentially related to anthropometric properties of the participant’s leg or brace fit.’ This was my main concern regarding this study, since only a small group of young and healthy participants was examined.
